# The incidence of deep vein thrombosis after anterior cruciate ligament reconstruction: An analysis using routine ultrasonography of 260 patients

**Yong Bum Joo[1], Young Mo Kim[1], Ju-Ho Song[2]\*, Byung Kuk An[1], Yun Ki Kim[1], Soon Tae Kwon[3]**

**1** Department of Orthopedic Surgery, Chungnam National University Hospital, Chungnam National University College of Medicine, Daejeon, Republic of Korea, **2** Department of Orthopedic Surgery, Chungnam National University Sejong Hospital, Chungnam National University College of Medicine, Sejong, Republic of Korea, **3** Department of Radiology, Chungnam National University Hospital, Chungnam National University College of Medicine, Daejeon, Republic of Korea

\* skypillar0221@gmail.com

**Data Availability Statement:** All relevant data are within the paper and its Supporting Information files.

## Abstract

### Background

Recent studies regarding deep vein thrombosis (DVT) after anterior cruciate ligament (ACL) reconstruction investigated only symptomatic complications. The purpose of this study was to assess the true incidence of DVT after ACL reconstruction, regardless of symptom manifestation.

### Materials and methods

Medical records of 260 patients who underwent isolated ACL reconstruction between January 2014 and December 2019 were retrospectively reviewed. Regardless of symptom manifestation, DVT was examined for all patients at 1 week postoperatively using ultrasonography. Demographics, injury mechanism (high energy direct injury and low energy indirect injury), soft tissue injury, preoperative anterior laxity, tourniquet time, and surgical technique (transtibial, anteromedial portal, and outside-in techniques) were investigated. Soft tissue injury was evaluated on magnetic resonance imaging (MRI) scans, based on the Tscherne classification. Risk factors for proximal DVT were identified using logistic regression analyses.

### Results

A total of 21 (8.1%) patients showed DVT. 5 (1.9%) patients had thrombosis at the popliteal vein; however, none of them exhibited symptoms. The other 16 patients had thrombosis at the distal veins: 1 patient at the anterior tibial vein, 5 patients at the posterior tibial vein, 3 patients at the peroneal vein, 6 patients at the soleal vein, and 1 patient at the muscular branch vein. The risk factors for proximal DVT included high energy direct injury (p = 0.013, odds ratio = 10.62) and grade 2 soft tissue injury (p = 0.039, odds ratio = 6.78).

**Funding:** This work was supported by research fund of Chungnam National University. The funders had no role in study design, data collection and analysis, decision to publish, or preparation of the manuscript.

**Competing interests:** The authors have declared that no competing interests exist.

## Conclusions

The true incidence of DVT, including symptomatic and asymptomatic complications, were 8.1% after ACL reconstruction. This rate is higher than the previously known incidence which has been investigated only for symptomatic patients. Injury mechanism and soft tissue injury should be assessed when considering thromboprophylaxis.

## Introduction

Deep vein thrombosis (DVT) has received great attention in the orthopedic society because extremity surgery is a well-known risk factor of DVT. Knee arthroscopy, although being a minimally invasive procedure, is also associated with DVT, and the incidence ranged from 0.4% to 17.9% [1–3]. This wide range of the incidence results from the heterogeneity of arthroscopic procedures and various diagnostic tools applied to detect DVT [4].

Anterior cruciate ligament (ACL) rupture is a result of trauma which is an important risk factor for DVT [5–7]. Recent studies reported low incidences of DVT after ACL reconstruction. Traven et al. found that venous thromboembolism occurred in 1.1% of the patients undergoing ACL reconstruction [4]. However, asymptomatic DVT was omitted because their study was based on a claim database. Schmitz et al. could not include asymptomatic DVT either in their study utilizing Swedish Knee Ligament Register database [8]. They recommended against the routine use of thromboprophylaxis, reporting that the incidence of DVT was 0.4%.

Asymptomatic DVT should be carefully observed because even distal DVT can develop into proximal DVT or pulmonary embolism [9, 10]. Moreover, symptomatic DVT has been defined differently in previous studies [11], and the previously known incidence would indicate only the tip of the iceberg [12]. The present study aimed to investigate the true incidence of DVT after ACL reconstruction using routine ultrasonography, regardless of symptom manifestation.

## Materials and methods

Medical records of 264 patients who underwent isolated ACL reconstruction between January 2014 and December 2019 were retrospectively reviewed after approval was obtained from institutional review board of Chungnam National University Hospital (No.2020-12-084). The requirement of written consent was waived for the retrospective review. Three patients who were already having anticoagulation therapy and one patient who had a history of DVT were excluded because the current guidelines already recommended thromboprophylaxis for those patients [13]. Accordingly, 260 patients were included in the study. All ACL reconstructions were performed using the transtibial, anteromedial portal, or outside-in techniques. Thromboprophylaxis was not routinely applied except for anti-embolism stockings and early postoperative ambulation.

### Evaluation of DVT and study design

DVT was examined for all patients at 1 week postoperatively. Lower extremity ultrasonography (Philips HD15, Bothwell, WA, USA) was performed by two experienced radiologists. If DVT was detected, computed tomography pulmonary angiography was considered after consultation with pulmonologists.

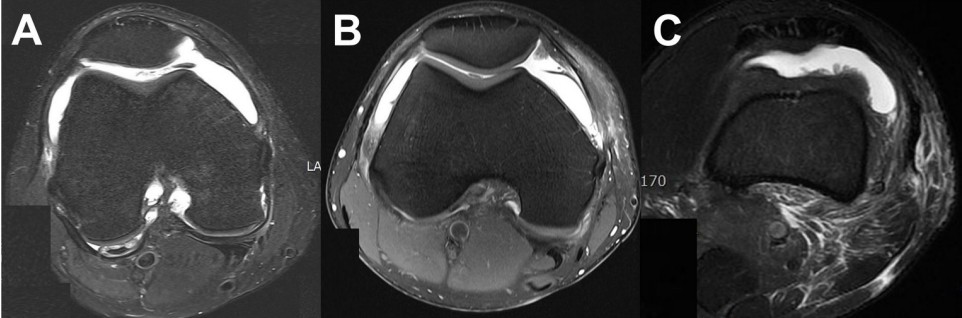

**Fig 1. The severity of soft tissue injury.** (A) Grade 0, none to minimal injury. (B) Grade 1, injury superficial to fascia. (C) Grade 2, injury deep to fascia.

The following factors were investigated: age, sex body mass index (BMI), smoking, hypertension, diabetes mellitus, injury mechanism (high energy direct injury and low energy indirect injury), soft tissue injury, preoperative anterior laxity, tourniquet time, and surgical technique (transtibial, anteromedial portal, and outside-in techniques). Soft tissue injury was evaluated on magnetic resonance imaging (MRI) scans, based on the Tscherne classification: [14] grade 0, none to minimal injury; grade 1, injury superficial to fascia; and grade 2, injury deep to fascia (Fig 1). Three orthopedic surgeons independently assessed soft tissue injury and all disagreements were resolved by discussion. Preoperative anterior laxity was measured as the side-to-side difference of anterior translation on stress radiographs.

Each factor was compared between the patients with DVT (DVT group) and those without DVT (non-DVT group). Risk factors for proximal DVT were identified using a multivariate logistic regression analysis.

## Statistical analysis

The sample size of each group (260 patients) was confirmed by post hoc analysis to achieve a power of 94.0% to reject the null hypothesis, with a significance level of 0.05. Post hoc power analysis was performed using G*Power (Version 3.1.7, Franz Faul, Christian-Albrechts-Universitätzu Kiel). Categorical variables including the incidence of DVT were analyzed by Chi-square test when the expected value of the cell was 5 or more in at least 80% of the cells; otherwise, Fisher exact test was used. Continuous variables were analyzed by $t$ test. All statistical analyses were performed using R software version 4.1.1 (R foundation for Statistical Computing, Vienna, Austria), with a p value $<$0.05 considered statistically significant.

## Results

Of 260 patients with a mean age of 29.9 ± 12.9 years (range, 13–60 years), DVT was noted in 21 (8.1%) patients. 5 (1.9%) patients had thrombosis at the popliteal vein; however, none of them exhibited symptoms. Computed tomography pulmonary angiography was performed in those 5 patients and there was no pulmonary embolism. The other 16 patients had thrombosis at the distal veins: 1 patient at the anterior tibial vein, 5 patients at the posterior tibial vein, 3 patients at the peroneal vein, 6 patients at the soleal vein, and 1 patient at the muscular branch vein.

The mean age of the patients in the DVT group and the non-DVT group were 35.1 ± 12.0 years and 29.5 ± 12.9 years, respectively (p = 0.012). 10 (47.6%) patients were smokers in the DVT group and 50 (20.9%) patients were in the non-DVT group (p = 0.012). 5 (23.8%) patients in the DVT group and 12 (5.0%) patients in the non-DVT group had high energy

**Table 1. Patient characteristics between the DVT and the non-DVT groups.**

| | Overall | Routine ultrasonography | | P value |
|---|---|---|---|---|
| | | DVT group (N = 21) | Non-DVT group (N = 239) | |
| Age, year | 29.9 ± 12.9 | 35.1 ± 12.0 | 29.5 ± 12.9 | 0.048 |
| Male / Female, n | 201/59 | 18/3 | 183/56 | 0.421 |
| BMI, kg/m$^2$ | 26.2 ± 5.1 | 25.8 ± 3.0 | 26.3 ± 6.4 | 0.827 |
| Smoking, n | 60 | 10 | 50 | 0.012 |
| Hypertension, n | 13 | 1 | 12 | 0.958 |
| Diabetes mellitus, n | 6 | 1 | 5 | 0.400 |
| Injury mechanism | | | | 0.007 |
| High energy direct, n | 17 | 5 | 12 | |
| Low energy indirect, n | 243 | 16 | 227 | |
| Soft tissue injury, n grade 0 / 1 / 2 / 3[b] | 113 / 98 / 49 / 0 | 4 / 8 / 9 / 0 | 109 / 90 / 40 / 0 | 0.009 |
| Preoperative anterior laxity, mm | 6.3 ± 3.7 | 7.3 ± 4.8 | 6.2 ± 3.6 | 0.217 |
| Tourniquet time, minute | 99.0 ± 17.1 | 102.8 ± 6.6 | 98.6 ± 17.7 | 0.283 |
| Surgical technique, n transtibial / AM portal / OI | 48 / 80 / 132 | 5 / 7 / 9 | 43 / 73 / 123 | 0.722 |

DVT, deep vein thrombosis; BMI, body mass index; AM portal, anteromedial portal; OI, outside in

[a]Data are reported as mean ± SD unless otherwise indicated.

[b]Soft tissue injury is graded based on the Tscherne classification: grade 0, none to minimal injury; grade 1, injury superficial to fascia; and grade 2, injury deep to fascia.

direct injury, which showed a significant difference between the groups (p = 0.007). Intergroup comparison also showed a difference in soft tissue injury (p = 0.009; Table 1). The logistic regression analysis found that high energy direct injury (p = 0.013, odds ratio = 10.62) and grade 2 soft tissue injury (p = 0.039, odds ratio = 6.78) were risk factors for proximal DVT (Table 2).

## Discussion

The most important finding of the present study was that the true incidence of DVT was higher (8.1%) than expected when DVT was examined irrespective of symptom manifestation. Ultrasonography identified 5 (1.9%) cases of proximal DVT at the popliteal vein which did not cause any symptoms. High energy direct injury and Tscherne grade 2 soft tissue injury were risk factors for proximal DVT in isolated ACL reconstruction; therefore, thromboprophylaxis should be considered after assessing these factors.

Previous studies have reported a wide range of DVT incidence, depending on whether asymptomatic DVT was included or not. A recent notable study by Schmitz et al. reported that the incidence of venous thromboembolism was 0.4% after ACL reconstruction [8]. Because their study was based on large registry database, only symptomatic complications could be investigated. However, Struijk-Mulder et al. found a 13% incidence of DVT using ultrasonography: 9% of asymptomatic DVT and 4% of symptomatic DVT [15]. Sun et al. also confirmed a high percentage of silent DVT after knee arthroscopic surgery in their study using venography [16]. Given that the natural history of asymptomatic DVT is still unclear [11], thromboprophylaxis protocols in ACL reconstruction should be established in consideration of asymptomatic DVT.

Asymptomatic DVT is not always benign. Even isolated distal DVT is associated with subsequent proximal DVT or pulmonary embolism [17, 18]. Brateanu et al. reported that 30 (7%)

**Table 2. Logistic regression analysis for proximal DVT after ACL reconstruction.**

| | *P* value | Exp(β) | 95% Confidence interval | |
|---|---|---|---|---|
| | | | Lower | Upper |
| Age | 0.633 | 1.03 | 0.95 | 1.14 |
| Sex | 0.298 | 1.16 | 0.01 | 5.06 |
| BMI | 0.771 | 1.05 | 0.74 | 1.49 |
| Smoking | 0.473 | 2.99 | 0.15 | 59.16 |
| Hypertension | 0.162 | 5.04 | 0.52 | 48.64 |
| Diabetes mellitus | 0.998 | NA | NA | NA |
| High energy direct injury[a] | 0.013 | 10.62 | 1.65 | 68.49 |
| Soft tissue injury[b] | | | | |
| Grade 1 | 0.920 | 1.10 | 0.18 | 6.68 |
| Grade 2 | 0.039 | 6.78 | 1.10 | 41.75 |
| Preoperative anterior laxity | 0.907 | 0.98 | 0.73 | 1.33 |
| Tourniquet time | 0.446 | 1.04 | 0.94 | 1.14 |
| Surgical technique | | | | |
| AM portal | 0.573 | 0.40 | 0.02 | 9.55 |
| OI | 0.070 | 0.04 | 0.01 | 1.29 |

DVT, deep vein thrombosis; ACL, anterior cruciate ligament; BMI, body mass index; AM portal, anteromedial portal; OI, outside in; NA, not applicable

[a]High energy direct injury was analyzed on the basis of low energy indirect injury.

[b]Grade 1 and grade 2 soft tissue injury were analyzed on the basis of grade 0 soft tissue injury.

of 450 patients with distal DVT had extension of thrombus to proximal veins within three months [10]. They created a model predicting the probability of developing proximal DVT or pulmonary embolism, showing that inpatient status and age ≥60 years were risk factors. Shimabukuro et al. also found consistent results in their recent study [11]. They recommended follow-up examinations for possible proximal extension of thrombus although most of the thrombi regressed without anticoagulation therapy. Thus, risk stratification is important when considering thromboprophylaxis in ACL reconstruction.

The current guidelines from the American College of Chest Physicians do not support thromboprophylaxis in knee arthroscopic surgery unless patients have previous venous thromboembolism [13]. However, the National Institute for Health and Clinical Excellence recommends 14 days of low-molecular-weight heparin when the total anesthesia time is longer than 90 minutes or the individual risk for venous thromboembolism outweighs the risk of bleeding [19]. When it comes to ACL reconstruction, Zhu et al. recently proved the efficacy of low-molecular-weight heparin in preventing venous thromboembolism [20]. Thromboprophylaxis protocols need to be focused on ACL reconstruction which usually takes more time than simple arthroscopic procedures such as meniscectomy.

Several previous studies have proved that smoking was a risk factor for DVT after ACL reconstruction. In their study using a national insurance database, Cancienne et al. compared the incidence of venous thromboembolism between the two matched groups which was divided according to tobacco use. They concluded that smoking was associated with venous thromboembolism as well as postoperative infection [21]. Two other studies based on large database also showed that smoking increased the odds of DVT [4, 22], and the present study found that the odds ratio was 3.2 (p = 0.022). However, smoking has not been included in current DVT risk assessment scoring systems [21, 23]. Injury mechanism and the severity of soft

tissue injury could not have been investigated in studies using large database. In this study, the odds ratios of high energy direct injury and grade 2 soft tissue injury were even higher (7.0 and 6.1, respectively) than that of smoking. Thromboprophylaxis protocols in ACL reconstruction should take these factors into account as well.

The mean age of the study population in this study was 29.9 ± 12.9 years, which was higher than that of cohorts included in previous studies regarding ACL reconstruction. In the register-based study by Schmitz et al., the mean age was 26.8 years [8]. In another large database study, Traven et al. found that the median age in those experiencing DVT was 32 years whereas the median age in those not experiencing DVT was 28 [4]. The risk of selection bias might also exist in smoking rate. The overall smoking rate was 23.1% in this study whereas it was 3.7%–6.0% in previous studies [4, 8]. Because smoking is an important risk factor for DVT [21, 22], selection bias should be considered when interpreting the results of the present study.Several limitations should be noted. First, the retrospective nature of this study could cause potential bias. Second, age was not a risk factor in a multivariate regression analysis, only being significant in univariate analyses. However, previous studies have found that age was associated with DVT [4, 8, 22]. The different results might be due to insufficient statistical power. Third, DVT might have occurred after ultrasonographic examination that was performed one week after ACL reconstruction.

## Conclusions

The true incidence of DVT, including symptomatic and asymptomatic complications, were 8.1% after ACL reconstruction. This rate is higher than the previously known incidence which has been investigated only for symptomatic patients. Injury mechanism and soft tissue injury should be assessed when considering thromboprophylaxis.

## Supporting information

**S1 Dataset.**
(CSV)

## Author Contributions

**Conceptualization:** Ju-Ho Song.

**Investigation:** Byung Kuk An, Yun Ki Kim, Soon Tae Kwon.

**Supervision:** Young Mo Kim.

**Writing – original draft:** Ju-Ho Song.

**Writing – review & editing:** Yong Bum Joo, Ju-Ho Song.

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
