## [Decision Letter · Decision Letter 0]

4 Oct 2022

PONE-D-22-21802The true incidence of deep vein thrombosis after anterior cruciate ligament reconstructionPLOS ONE

Dear Dr. Song,

Thank you for submitting your manuscript to PLOS ONE. After careful consideration, we feel that it has merit but does not fully meet PLOS ONE’s publication criteria as it currently stands. Therefore, we invite you to submit a revised version of the manuscript that addresses the points raised during the review process.

First of all, I would like to congratulate you for this intersting article entitled 'The true incidence of deep vein thrombosis after anterior cruciate ligament reconstruction'.

The manuscript has been evaluated by three reviewers, and their comments are available below.

I also want to ask you a few questions:

-Do you routinely have ultrasound done to each of your cases after this type of surgical repair?

- Have you administered prophylactic anticoagulation to these patients according to your National Guidelines?

- Were all DVT patients or those with popliteal DVT asymptomatic?

-Why did you perform thoracic CT for pulmonary embolism (PE) in all patients with DVT? - How many patients have you detected PE? You should also add this information to the results section.

We look forward to receiving your revised manuscript.

Kind regards,

Eyüp Serhat Çalık

Academic Editor

PLOS ONE

Reviewers' comments:

Reviewer's Responses to Questions

**Comments to the Author**

1. Is the manuscript technically sound, and do the data support the conclusions?

Reviewer #1: Partly

Reviewer #2: Partly

Reviewer #3: Partly

2. Has the statistical analysis been performed appropriately and rigorously? 

Reviewer #1: Yes

Reviewer #2: I Don't Know

Reviewer #3: Yes

3. Have the authors made all data underlying the findings in their manuscript fully available?

Reviewer #1: Yes

Reviewer #2: Yes

Reviewer #3: Yes

4. Is the manuscript presented in an intelligible fashion and written in standard English?

Reviewer #1: Yes

Reviewer #2: Yes

Reviewer #3: Yes

5. Review Comments to the Author

Reviewer #1: The authors present a well written manuscript but their study has a number of significant flaws which make it unacceptable for publication. First and foremost, the study is based on a single timepoint ultrasound evaluation for venous thrombosis one week after ACL reconstruction. There is no clinical application or utility of this finding. Without reassessment at a later timepoint with at least clinical information, and preferably repeat ultrasound evaluation, the finding of venous thrombosis in 8.1% (6.2% distal to the popliteal vein) based on US one week after surgery is an incidental finding. Unfortunately, it raises the proverbial question, so what? This finding is not surprising or concerning and will not change practice. Ideally, in order to have a clinically relevant study worthy of publication, the patients would have been re-evaluated serially, at least at 1, 3 and 6 months after surgery, with US to determine the natural history of these otherwise incidental findings. The suspicion is that the vast majority will resolve without any clinical manifestation. There is no indication for therapeutic treatment of the distal thromboses and it is not clear if the asymptomatic popliteal vein thromboses should be treated.

Specific comments

Lines 71-3: Delete this sentence

Lines 97-98: Was tourniquet used during these surgeries? If so, what was the average tourniquet time?

Lines 106-7: Distal DVT is not considered clinically important. Without serial repeat imaging, these should be excluded.

Lines 107-8: CT pulmonary angiography was performed for distal DVT? Did anyone have a PE? The CT angio results should be reported.

Lines 109-113: Why not tourniquet time?

Lines 121-2: Recommend regression for proximal DVT only.

This cohort is atypical for ACL reconstruction as it is older, more male and more heavily smoking than most ACL reconstruction cohorts. This may bias the results and should be explored at length in the discussion.

Reviewer #2: Deep vein thrombosis is an important issue that is frequently encountered after orthopedic surgical interventions. In particular, screening of the asymptomatic patient group has a very important place in this regard.

In this study, I have some questions to the authors.

1- It is stated that it is a retrospective study. Can you explain the fact that in a retrospective study, control USG was requested one week after the surgical intervention? Is it a routine practice to perform control USG on all patients?

2- Identification of risk factors for DVT was stated as the aim of the study. However, the data on this subject is limited in the discussion and findings section. Only one analysis was made on smoking. I think it would be appropriate to change the purpose of the study or to discuss other risk factors in the discussion section.

Reviewer #3: This manuscript is the very interesting research about incidence of DVT after ACL reconstruction. The incidence of DVT after orthopaedic surgery is a serious problem, and it is important to examine the incidence of DVT after ACL reconstruction. I think this paper is a good paper, but there is a problem that it does not confirm that DVT did not occur before surgery. The ACL injury is a traumatic injury and may have been concurrent with deep vein injury at the time of injury, and DVT may have occurred preoperatively. I consider that the high incidence of postoperative DVT in multiligament injuries is also due to the high-energy injury at the time of injury, and the possibility that DVT had already occurred before surgery cannot be ruled out. We believe it is necessary to make an accurate statement on this point. These points should be revised for this study to be relevant for publication in the PLOS ONE.

Title

For the aforementioned reasons, I consider the expression "True" to be an overstatement. Please change the title.

Introduction, Materials and Methods, Results, Figures

Well written.

6. PLOS authors have the option to publish the peer review history of their article (what does this mean?). If published, this will include your full peer review and any attached files.

Reviewer #1: No

Reviewer #2: **Yes: **FERHAT BORULU

Reviewer #3: No

---

## [Author Response · Author response to Decision Letter 0]

27 Oct 2022

First of all, I would like to congratulate you for this intersting article entitled 'The true incidence of deep vein thrombosis after anterior cruciate ligament reconstruction'.

The manuscript has been evaluated by three reviewers, and their comments are available below.

I also want to ask you a few questions:

- Do you routinely have ultrasound done to each of your cases after this type of surgical repair?

Ultrasonography at one week postoperatively was routinely performed during the study period. However, the postoperative protocol has now been changed based on the results of this study. 

- Have you administered prophylactic anticoagulation to these patients according to your National Guidelines?

Thromboprophylaxis was not routinely applied except for anti-embolism stockings and early postoperative ambulation (Line 100–101) because the current guidelines did not advocate for routine anticoagulation in knee arthroscopy.

- Were all DVT patients or those with popliteal DVT asymptomatic?

All popliteal DVT patients had no related symptoms (Line 151).

-Why did you perform thoracic CT for pulmonary embolism (PE) in all patients with DVT? - How many patients have you detected PE? You should also add this information to the results section.

CT for pulmonary embolism was performed only for the patients with proximal DVT, which was determined after consultation with pulmonologists. There was no pulmonary embolism detected on CT scans. The manuscript has been revised accordingly (Line 108–110, 150–151). 

Reviewers' comments:

Reviewer's Responses to Questions

Comments to the Author

1. Is the manuscript technically sound, and do the data support the conclusions?

Reviewer #1: Partly

Reviewer #2: Partly

Reviewer #3: Partly

2. Has the statistical analysis been performed appropriately and rigorously?

Reviewer #1: Yes

Reviewer #2: I Don't Know

Reviewer #3: Yes

3. Have the authors made all data underlying the findings in their manuscript fully available?

Reviewer #1: Yes

Reviewer #2: Yes

Reviewer #3: Yes

4. Is the manuscript presented in an intelligible fashion and written in standard English?

Reviewer #1: Yes

Reviewer #2: Yes

Reviewer #3: Yes

5. Review Comments to the Author

Reviewer #1: The authors present a well written manuscript but their study has a number of significant flaws which make it unacceptable for publication. First and foremost, the study is based on a single timepoint ultrasound evaluation for venous thrombosis one week after ACL reconstruction. There is no clinical application or utility of this finding. Without reassessment at a later timepoint with at least clinical information, and preferably repeat ultrasound evaluation, the finding of venous thrombosis in 8.1% (6.2% distal to the popliteal vein) based on US one week after surgery is an incidental finding. Unfortunately, it raises the proverbial question, so what? This finding is not surprising or concerning and will not change practice. Ideally, in order to have a clinically relevant study worthy of publication, the patients would have been re-evaluated serially, at least at 1, 3 and 6 months after surgery, with US to determine the natural history of these otherwise incidental findings. The suspicion is that the vast majority will resolve without any clinical manifestation. There is no indication for therapeutic treatment of the distal thromboses and it is not clear if the asymptomatic popliteal vein thromboses should be treated.

Specific comments

Lines 71-3: Delete this sentence

The manuscript has been revised accordingly. 

Lines 97-98: Was tourniquet used during these surgeries? If so, what was the average tourniquet time?

A tourniquet was routinely used and tourniquet time was presented instead of surgical time, as suggested by another reviewer (Line 36, 114). The average surgical time was 99.0 ± 17.1 minutes (Table 1). 

Lines 106-7: Distal DVT is not considered clinically important. Without serial repeat imaging, these should be excluded.

The manuscript has been revised accordingly (Line 106–108).

Lines 107-8: CT pulmonary angiography was performed for distal DVT? Did anyone have a PE? The CT angio results should be reported.

CT for pulmonary embolism was performed only for the patients with proximal DVT, which was determined after consultation with pulmonologists. There was no pulmonary embolism detected on CT scans. The manuscript has been revised accordingly (Line 108–110, 150–151). 

Lines 109-113: Why not tourniquet time?

As suggested, tourniquet time was presented instead of surgical time (Line 36, 114).

Lines 121-2: Recommend regression for proximal DVT only.

As suggested, the design of the study has been revised, focusing on proximal DVT only (Line 38, 123–125, 161–163, Table 2). Accordingly, the results and conclusion have been slightly changed (Line 44–46, 49, 161–163, 176–177, 238). 

This cohort is atypical for ACL reconstruction as it is older, more male and more heavily smoking than most ACL reconstruction cohorts. This may bias the results and should be explored at length in the discussion.

The discussion has been revised accordingly (Line 219–227).

Reviewer #2: Deep vein thrombosis is an important issue that is frequently encountered after orthopedic surgical interventions. In particular, screening of the asymptomatic patient group has a very important place in this regard.

In this study, I have some questions to the authors.

1- It is stated that it is a retrospective study. Can you explain the fact that in a retrospective study, control USG was requested one week after the surgical intervention? Is it a routine practice to perform control USG on all patients?

Ultrasonography at one week postoperatively was routinely performed during the study period. However, the postoperative protocol has now been changed based on the results of this study. 

2- Identification of risk factors for DVT was stated as the aim of the study. However, the data on this subject is limited in the discussion and findings section. Only one analysis was made on smoking. I think it would be appropriate to change the purpose of the study or to discuss other risk factors in the discussion section.

As suggested, the purpose of the study has been revised (Line 30, 88).

Reviewer #3: This manuscript is the very interesting research about incidence of DVT after ACL reconstruction. The incidence of DVT after orthopaedic surgery is a serious problem, and it is important to examine the incidence of DVT after ACL reconstruction. I think this paper is a good paper, but there is a problem that it does not confirm that DVT did not occur before surgery. The ACL injury is a traumatic injury and may have been concurrent with deep vein injury at the time of injury, and DVT may have occurred preoperatively. I consider that the high incidence of postoperative DVT in multiligament injuries is also due to the high-energy injury at the time of injury, and the possibility that DVT had already occurred before surgery cannot be ruled out. We believe it is necessary to make an accurate statement on this point. These points should be revised for this study to be relevant for publication in the PLOS ONE.

Title

For the aforementioned reasons, I consider the expression "True" to be an overstatement. Please change the title.

As suggested, ‘true’ was removed and the title has been changed to ‘The incidence of deep vein thrombosis after anterior cruciate ligament reconstruction: an analysis using routine ultrasonography of 260 patients’.

Introduction, Materials and Methods, Results, Figures

Well written.

---

## [Decision Letter · Decision Letter 1]

24 Nov 2022

PONE-D-22-21802R1The incidence of deep vein thrombosis after anterior cruciate ligament reconstruction: an analysis using routine ultrasonography of 260 patientsPLOS ONE

Dear Dr. Song,

Thank you for submitting your manuscript to PLOS ONE. After careful consideration, we feel that it has merit but does not fully meet PLOS ONE’s publication criteria as it currently stands. Therefore, we invite you to submit a revised version of the manuscript that addresses the points raised during the review process.

We look forward to receiving your revised manuscript.

Kind regards,

Eyüp Serhat Çalık

Academic Editor

PLOS ONE

Journal Requirements:

Additional Editor Comments:

Dear Authors

We reviewed the revised version of your article and your responses to the reviewers. Your manuscript has been additionally evaluated by one more reviewer. We request that you re-upload your work, which you will edit in accordance with the recommendations of Reviewer 4, as soon as possible.

Reviewers' comments:

Reviewer's Responses to Questions

**Comments to the Author**

1. If the authors have adequately addressed your comments raised in a previous round of review and you feel that this manuscript is now acceptable for publication, you may indicate that here to bypass the “Comments to the Author” section, enter your conflict of interest statement in the “Confidential to Editor” section, and submit your "Accept" recommendation.

Reviewer #4: (No Response)

2. Is the manuscript technically sound, and do the data support the conclusions?

Reviewer #4: (No Response)

3. Has the statistical analysis been performed appropriately and rigorously? 

Reviewer #4: (No Response)

4. Have the authors made all data underlying the findings in their manuscript fully available?

Reviewer #4: (No Response)

5. Is the manuscript presented in an intelligible fashion and written in standard English?

Reviewer #4: (No Response)

6. Review Comments to the Author

Reviewer #4: The purpose of this paper was to assess the incidence of DVT after ACL reconstruction, regardless of symptom manifestation, by analyzing 260 patients. They reported the incidence of DVT to be 8.1%, which is higher than the previously known incidence.

1. Some values in Table 2 should be checked. For example,

a. The lower bound of CI for age = 1.0 which is the same as original exp(beta)?

b. CI for diabetes mellitus

c. Lower bound of CI for OI.

7. PLOS authors have the option to publish the peer review history of their article (what does this mean?). If published, this will include your full peer review and any attached files.

Reviewer #4: No

---

## [Author Response · Author response to Decision Letter 1]

25 Nov 2022

Reviewer #4: The purpose of this paper was to assess the incidence of DVT after ACL reconstruction, regardless of symptom manifestation, by analyzing 260 patients. They reported the incidence of DVT to be 8.1%, which is higher than the previously known incidence.

1. Some values in Table 2 should be checked. For example,

a. The lower bound of CI for age = 1.0 which is the same as original exp(beta)?

b. CI for diabetes mellitus

c. Lower bound of CI for OI.

As suggested, 95% CI has been presented to two decimal places to avoid confusion. The manuscript has also been revised accordingly. 

Regarding the CI of diabetes mellitus, the values have been replaced with NA (not applicable).

---

## [Editor Report · Decision Letter 2]

1 Dec 2022

The incidence of deep vein thrombosis after anterior cruciate ligament reconstruction: an analysis using routine ultrasonography of 260 patients

PONE-D-22-21802R2

Dear Dr. Song,

We’re pleased to inform you that your manuscript has been judged scientifically suitable for publication and will be formally accepted for publication once it meets all outstanding technical requirements.

Kind regards,

Eyüp Serhat Çalık

Academic Editor

PLOS ONE
---

## [Editor Report · Acceptance letter]

5 Dec 2022

PONE-D-22-21802R2 

The incidence of deep vein thrombosis after anterior cruciate ligament reconstruction: an analysis using routine ultrasonography of 260 patients 

Dear Dr. Song:

I'm pleased to inform you that your manuscript has been deemed suitable for publication in PLOS ONE. Congratulations! Your manuscript is now with our production department. 

Kind regards, 

on behalf of

Dr. Eyüp Serhat Çalık 

Academic Editor

PLOS ONE